# Artificial Neural Network Model to Predict Final Construction Contract Duration

Abdullah M. Alsugair [ID], Khalid S. Al-Gahtani *[ID], Naif M. Alsanabani [ID], Abdulmajeed A. Alabduljabbar and Abdulmohsen S. Almohsen [ID]

Department of Civil Engineering, King Saud University, P.O. Box 2454, Riyadh 114, Saudi Arabia; amsugair@ksu.edu.sa (A.M.A.); nalsanabani@ksu.edu.sa (N.M.A.); 442105816@student.ksu.edu.sa (A.A.A.); asmohsen@ksu.edu.sa (A.S.A.)
* Correspondence: kgahtani@ksu.edu.sa

**Abstract:** Forecasting the final construction contract duration at an early stage plays a vital role in the progress of a project. An inaccurate project duration prediction may lead to the project's benefits being lost. It is essential to precisely predict the duration due to the presence of several different factors. This paper contributed to developing a model to predict final construction contract duration (*FCCD*) in the early stages based on parameters characterized as few and shared for any contract. (contract cost, contract duration, and sector). This paper developed an Artificial Neural Network (ANN) model based on 135 Saudi construction project data. The development model has three stages. The first stage was standardization and augmentation using Zavadskas and Turskis' logarithmic and Pasini methods. The second and third stages were the first and second analyses of the ANN models, respectively. The first analysis aimed to promote the used data and integrate them into the second analysis to develop the ANN model. The ANN models were compared with three linear regression (LR) models (LR1, LR2, and LR3) and other models in the literature. The results revealed that the accuracy of the ANN model provides reasonable accuracy with an average mean absolute percentage error (MAPE) of 12.22%, which is lower than the LR3's MAPE by 27.03%. The accuracy of the ANN model is similar to that of earned value management (EVM) in the previous study. This paper supports research to deal with relatively little data and integrate them into a neural network. The ANN model assists the stakeholder in making appropriate decisions for the project during the pre-tendering phase by predicting the actual contract duration based on the *CC*, *CD*, and project ector.

**Keywords:** duration; contract; logarithmic; transformed; cost; sector; errors

## 1. Introduction

The construction industry significantly contributes to countries' economic progress. Construction is an industry that contributes significantly to the overall Gross Domestic Product (GDP) and is expected to expand. Delays in construction projects have become a widespread issue due to the complexity of the construction industry. Despite having a positive impact on the economy and technological improvements in the sector [1,2], construction delays have a wide range of social and economic repercussions. These delays negatively impact sustainability's social, environmental, and financial triple bottom lines [3]. Delays can lead to schedule and cost overruns, decreased contractor earnings, additional losses for the owner's capital due to an extended construction phase, mistrust between the owner and contractor, legal battles involving many parties, and outright project abandonment. Gebrehiwet and Luo [4] noted that cost overruns, contract cancellation, arbitration, and litigation are some of a delay's crucial effects. According to Khattri et al. [5], a delay can result in disagreements, cost overruns, time overruns, abandonment, negotiation, legal action, litigation, and complete desertion. Numerous studies have been carried out over the years to address this significant issue, especially to identify the underlying factors that increase the probability of a building delay and its adverse effects.

Researchers in multiple nations have revealed numerous harmful consequences of delays. Hecker [6] claimed that significant cost overruns ranging from 40% to 400% occurred on various infrastructure projects in the United States of America. In the United Kingdom of Great Britain, just 38% of projects were finished within 5% of the contract timeline program, and only 70% were completed within 5% of the tender cost [7]. Additionally, according to [8], only one-eighth of Australian building projects were finished by the deadlines outlined in the contracts, and the typical schedule overrun was greater than 40%. Moreover, Sodangi and Salman [9] stated that about 70% of projects in Nigeria encountered delays, demonstrating that delays are a significant issue in Nigerian construction. Only 30% of building projects in the Kingdom of Saudi Arabia (KSA) were finished on time, while the typical time overrun was up to 30% [10].

Most studies have focused on identifying and analyzing the factors affecting delays in projects [4,11–13]. They pointed out that the factors that cause poor project performance still need to be fully understood. The causes of time delays were different from region to region. There is a need for more studies to predict the final construction contract duration in the early stages to assist stakeholders in deciding whether to continue or halt the project. Therefore, this paper aims to develop forecasting models for predicting the final construction duration project-based Artificial Neural Network (ANN) based on contract cost, contract duration, and sector type of 135 Saudi Arabia construction project data. Developing ANN models goes through three processes. This paper contributed to designing a model to forecast *FCCD* in the early stages based on parameters represented as few and shared for any contract. (*CC*, *CD*, and sector). The theoretical contribution of the paper is to utilize small data to develop the ANN model. In addition, the practical implications of the paper are to forecast the final construction duration at the pre-tendering stage.

## 2. Literature Review

The forecast construction contract duration studies have been distributed into three types of models. The first type of model is to predict the final duration based on the causes of the project delay. The second type of model is based on historical data, such as earned value management (EVM). The third type of model depends on the characteristics of a contract (artificial neural network, regression, hybrid models) or on information from the last completed contract that had the same condition as the required contract (case-based reasoning, CBS).

### 2.1. Forecast Model Based on the Causes of the Time Delays

In this section, the researchers tried to find the significant factors that affected the time delay and integrated them as input data to forecast the final construction contract duration. In a general project, Al-Gahtani et al. [14] used ten previously discovered criteria to construct a simulation forecasting model for the delay duration in Saudi projects using system dynamics. In order to consider the ten factors that influence project delay, they carried out a systematic, integrated approach using the DEMATEL methodology and system dynamics (SD). This work solved the challenge of methodically creating a causal loop diagram inside the SD modeling process using the DEMATEL technique. Next, consistency and extreme conditions were tested on the generated SD model. Then, it was implemented and validated using three case studies in KSA by contrasting each case study's real and fitted progress curves. In addition, Ajayi and Chinda [2] developed a model to examine the impact of the factors on the final construction project time. The model combined two mathematical decision-making techniques, DEMATEL and SD modeling. The simulation findings highlight the significance of avoiding design errors at the project's beginning (or preconstruction stage) to reduce project delays.

For highway projects, Pewdum et al. [15] evolved models to project the final cost and duration of a highway construction project while it was still in the planning stages. Before designing the forecasting models, project data were gathered and examined to determine the variables influencing the project's ultimate budget and duration. The research for these

models was based on the ANN. Han et al. [16] examined the influence of the non-value-adding effort generated from design errors and changes in design on the time delay of the project using system dynamics.

In order to facilitate reliable project delay risk analysis and forecasting using objective data sources, Gondia et al. [17] refined and built machine learning (ML) algorithms (decision tree and naïve Bayesian algorithms). As a result, the relevant delay risk sources and components were first found. A multivariate data set of past project timeliness and delay-inducing risk sources was assembled. Exploratory data analysis was then used to reveal the system's intricacy and interconnectivity. In order to anticipate the extent of project delays, the two appropriate algorithms were found and trained using the data set. These models used decision trees and naive Bayesian classification algorithms. Finally, cross-validation tests were performed on both models to assess their predictive abilities. The models were then contrasted using performance metrics pertinent to ML.

### 2.2. Forecast Model Based on Characteristics of a Contract or Project

Although the earned value management (EVM) approach is a successful project oversight and management strategy in terms of foretelling the cost performance index and other cost indicators, the technique may require more improvements to be more effective at estimating the project's completion time [18,19]. Vanhoucke and Vandevoorde [19] assumed that project activities and precedence relations were known to predict the final contract duration (*FCCD*). Urgilés et al. [20] examined the adequate EVM and value schedule to forecast the final duration of hydroelectric power generation projects. Sackey et al. [21] also developed a new method based on the EVM to forecast the final construction contract duration (*FCCD*). They used the actual time spent on each activity. One of the challenges faced by the users of the management method in predicting the actual duration of the contract is that the method requires historical data for the project. In other words, EVM also needs accurate information from a project, such as its cost, earned value, and planned value, at any given time, and it may not be possible to predict it at an early stage of the contract.

On the other hand, the case-based reasoning method is mainly used to forecast the construction project cost. However, Jin et al. [22] established a CBR model that can correctly predict the *FCCD* at the planning stage.

### 2.3. Regression and ANN Models

Several studies utilized regression and ANN models to estimate the *FCCD*. For example, Skitmore and Ng [23] developed a regression model based on cross-validation. The model parameters were project type, sector, contractor selection, and the 93 Australian building project model. Thomas and Thomas [24] developed a regression model to predict the building project duration based on 51 historical data. The model parameters were the area of the building, estimated duration, and estimated cost. The model did not consider the project sectors, and the model cannot be used for different types of projects, such as electrical or mechanical projects. The artificial neural network method proved more advanced and performed better than the regression model [25]. The ANN model developed by [26] was to forecast the duration of building projects. The input data included the number of floors, foundation type, activities, contractor class and client class, and floor area. The mean absolute error was 25.9%. Moreover, Gab-Allah et al. [27] established an ANN model for predicting the building project. The parameters were the type of clients, construction quality, project location, the total height of the project, client coordination with contractor staff, contract type (unit price contract/lump sum), contactor selection method, and quality of project documentation. The maximum error of the model was 20%.

The previous models required specific information, which varies from one project to another, such as the model developed by Al-Gahtani et al. [14], Ajayi and Chinda [2], Pewdum et al. [15], or contract data that should be available through the construction stage, such as Sackey et al. [23]. Although the CBR method has proven its effectiveness in predicting the duration of the contract, it requires the availability of a previously completed

project similar to the one required in terms of characteristics and operational conditions, which may be difficult to provide. In terms of the regression and ANN models, the above model was utilized for building project duration and cannot be generalized to other projects. Therefore, there is a need to develop a predictive model using the ANN model, which is used for different projects and is based on common and available data. This paper contributed to the development of a model to predict *FCCD* in the early stages based on a few common node criteria (*CC*, *CD*, and sector). Those parameters are characterized as being few and common for any contract. However, the ANN models need extensive data to provide a reliable and adequate forecast model. In this paper, the available data were relatively small (135), making it challenging to deal with them using an ANN model. To address the issue, the relatively small data issue was solved using the two methods that regard standardization (Zavadskas and Turskis' logarithmic) and augmentation data by utilizing the method introduced by [28], which was then utilized in developing the ANN model. The input data tested consists of contract cost, contract duration, and project sectors. This study presented a reasonable-accuracy prediction model based on KSA project data. The analysis approach used to create the model can also be applied to projects from other areas of the globe.

### 3. Artificial Neural Network (ANN) Model

This section provides an important and simple introduction to ANN model structures. The Artificial Neural Network (ANN) is an ML method that emerged from the concept of biological neural networks in the human brain. In situations where the actual process is complex and we are unsure of the nature of each phenomenon involved, it is one of the most excellent tools for value prediction [29]. Civil Engineering is one of the areas that benefits from ML. Such applications include earthquake engineering, structural health monitoring, damage identification and detection, and structural design. The VULMA ML-based tool is an example of such application automation, establishing a seismic vulnerability score for building structures [30,31].

A model known as an ANN builds an algorithm from any function to estimate the outcome [29]. The structures of the ANN consist of three layers: input layer, hidden layer (one or more than one), and hidden layer, as shown in Figure 1. The main aim of the hidden layer is to extract some of the most relevant patterns from the inputs and send them on to the next layer for further analysis. The mechanism of each hidden neuron consists of two sequence functions, *S* and *a*, as shown in Figure 2. The hidden layer also accelerates and improves the efficiency of the network by recognizing just the most essential information from the inputs and discarding redundant information. The benefit of using an ANN in Statistical Package for Social Sciences software is its simplicity and ability to handle small amounts of data, like the sector in this article. Moreover, the IBM SPSS Statistics 20 program can provide a neat illustration of using the ANN model with the strong relationships among neurons and the bias values. Additionally, SPSS makes it simple to choose the percentage of training and testing processes and provides the relative errors of the two processes together with the expected result values. Moreover, the program can provide the weighting values among the connections of the neurons and the output computed by the ANN model. On the other hand, the activation function allows the model to capture nonlinear relationships between the inputs. In addition, the activation function contributes to converting the input into a more usable output. The types of activation functions are hyperbolic tangent functions (Equation (1)), sigmoid functions (Equation (2)),

$$f = \frac{1 - e^{-2x}}{1 + e^{-2x}} \tag{1}$$

$$f = \frac{1}{1 + e^{-x}}$$ (2)

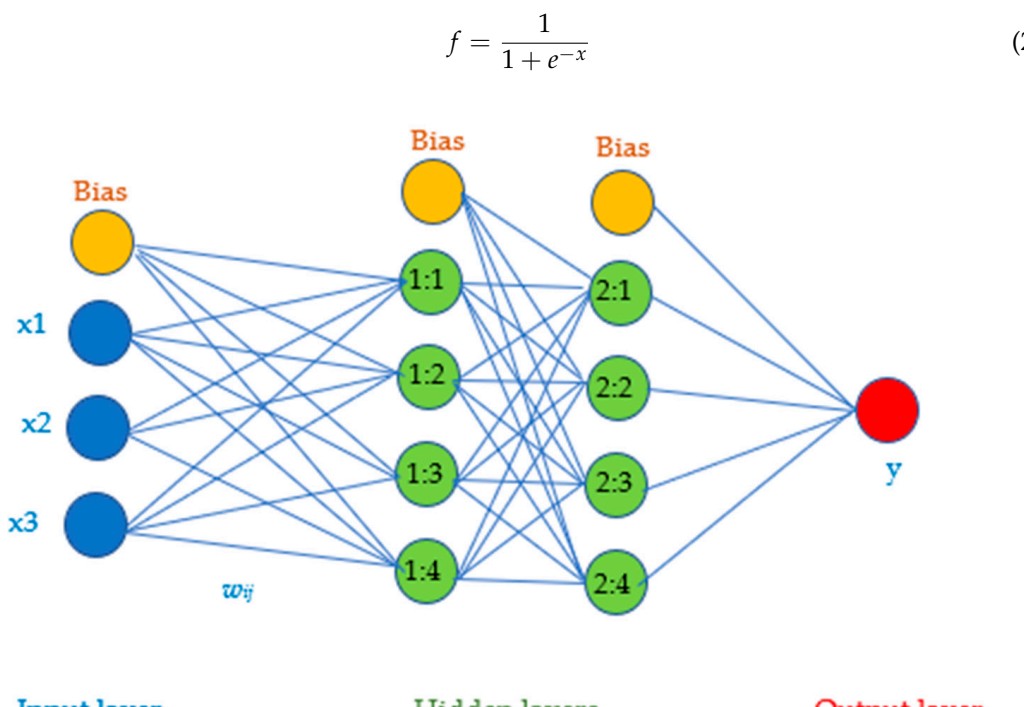

**Figure 1.** ANN structures.

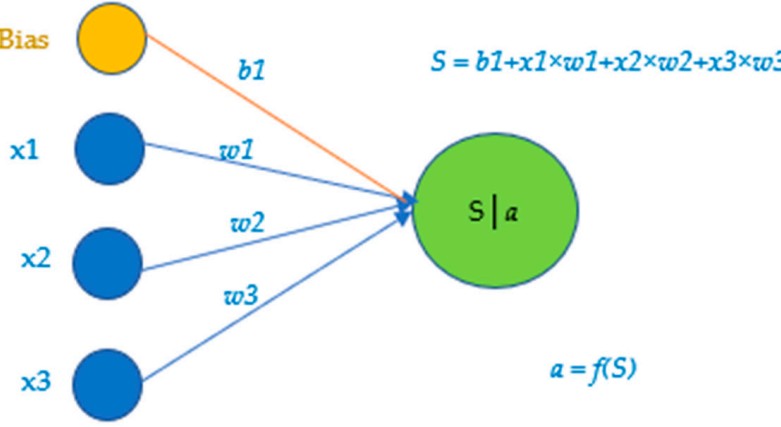

**Figure 2.** Illustrative example of hidden neuron work.

## 4. Methodology

The methodology is mainly comprised of three stages: data initialization, ANN development, and an evaluation model. The data initialization consists of data collection, standardization, and augmentation, while the ANN development includes the first and second analyses. The first analysis stage represented normalizing and maximizing data using Zavadskas and Turskis' logarithmic method and the method introduced by [32]. These methods overcome the issue of relatively small data. Then, the accuracy of the developed ANN model was determined using the mean absolute percentage error (MAPE). The second analysis stage is developing the ANN models on the modified data generated from the first stage, as shown in Figure 3. The ANN models were evaluated by comparing the results with several other past models in the literature.

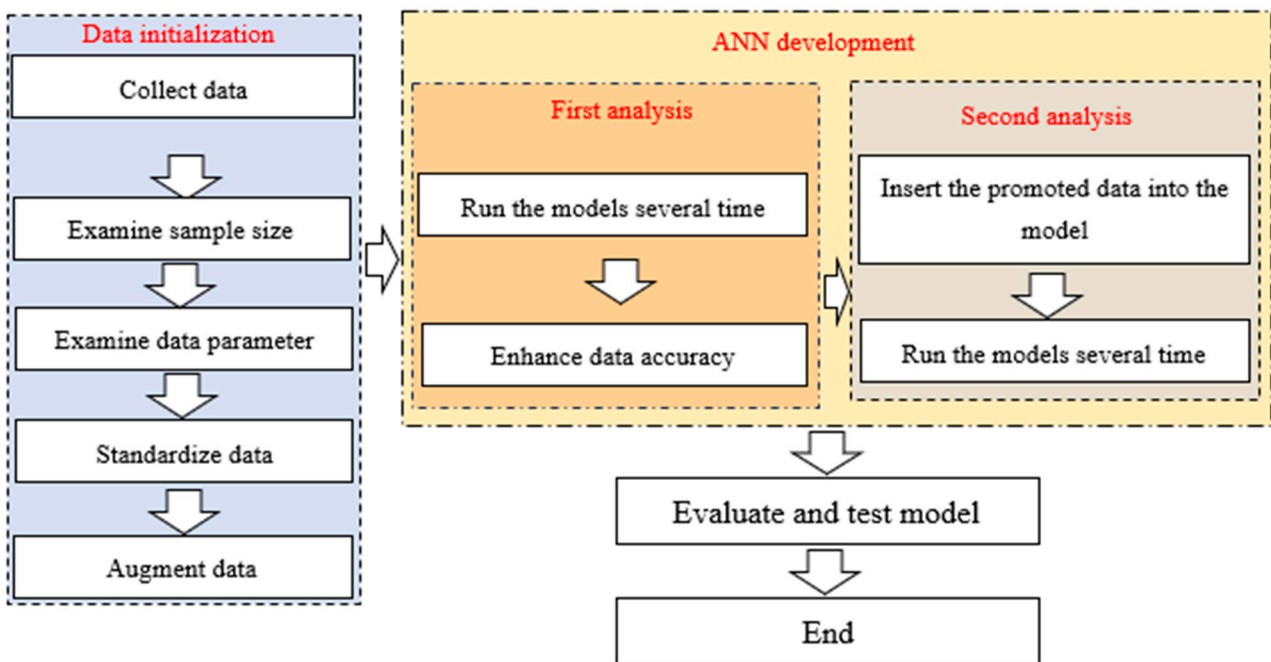

**Figure 3.** Methodology flow chart.

### *4.1. Data Initialization*

The section mainly aimed to collect and prepare data to make them suitable for use in artificial neural networks. It can be achieved by performing three steps: collecting, standardizing, and augmenting the standardized data. Gebrehiwet and Luo [4] and Khattri et al. [5] pointed out that the time delay had an influence on the cost and time overrun, which were reflected in the contract by *CC* and *CD*, respectively. Therefore, the collected data parameters were *CD*, *CC*, *FCCD*, contract sector (public, semi-public, private), and contract type (building, electric, mechanic).

### 4.1.1. Data Collection

The information from previously finished projects was required to create models forecasting the final construction duration. A survey was created and sent to numerous organizations to gather information from the finished building projects in KSA (Appendix A).

The project's scheduled start and finish dates were compared with the actual dates to determine the projected and real project durations. The data were collected, reviewed, and the invalid ones were eliminated. Models for predicting real construction duration were developed and validated using data from 135 projects completed in KSA. The frequencies of public, semi-public, and private were 80 (59.26%), 49 (36.30%), and 5 (4.44%), respectively. In addition, the *CC* ranges from 18,200 SAR to 650,000,000 SAR. On the other hand, the *CD* varied from 0.47 months to 138.30 months, while the *FCCD* changed from 0.37 months to 146.00 months. The frequencies of building, electric, and mechanic were 66 (48.89%), 14 (10.37%), and 55 (40.74%), respectively.

### 4.1.2. Sample Size Examine

The sample size of 135 projects can be examined by calculating the minimum size that follows the normal distribution using Equation (3), which is based on the confidence level (95%), probability value choice (*p*), which is set at 0.5, and confidence interval, which should be less than 0.2 [33].

$$Sample\ size = \frac{Z^2 p(1-p)}{C^2} \tag{3}$$

where $Z$ is a value corresponding to a 95% confidence level equal to 1.96. By setting the $C$ and $p$ to 0.10 and 0.50, respectively, the minimum sample was 49 projects, which was less than the collected projects (135). Therefore, the collected data was adequate.

### 4.1.3. Data Parameters Examination

This section aims to examine the impact of *CC*, *CD*, project sector, and project type on the *FCCD* by performing the correlation test between the *FCCD* and the other parameters. The results of the test are shown in Table 1. The *CD*, *CC*, and project sectors correlated with *FCCD* with Pearson coefficients of 0.784, 0.424, and 0.520, respectively. However, the project type did not correlate with *FCCD*, where the $p$-value of the test and Pearson coefficient were 0.666 (more than 0.05) and 0.037 (very weak correlation), respectively. Therefore, the parameters considered as the ANN model's input layer were *CD*, *CC*, and project sector (public, semi-public, and private).

**Table 1.** Correlation test results for different parameters.

|   | Parameters | | Pearson Correlation Coefficient | $p$-Value | N |
|---|---|---|---|---|---|
| 1 | *FCCD* | *CD* | 0.784 | <0.0001 | 135 |
| 2 | *FCCD* | *CC* | 0.424 | <0.0001 | 135 |
| 3 | *FCCD* | Project sector | 0.520 | <0.0001 | 135 |
| 4 | *FCCD* | Project types | 0.037 | 0.666 | 135 |

### 4.1.4. Data Standardization

Anysz et al. [28] stated that several approaches to standardizing input data for ANN result in varied values for accuracy metrics. They examine the six normalized methods (vector, Manhattan, maximum, Weitendorf's linear, Peldschus' nonlinear, Zavadskas and Turskis' logarithms, and Jüttler–Korth linear). Anysz et al. concluded that the Zavadskas and Turskis' logarithms provide tiny errors between the actual and computed output. Therefore, the method was considered to standardize the 135 data sets. The standardized formula is shown in Equation (4).

$$\overline{x}_i = \frac{\ln x_i}{\ln \prod_{i=1}^{n} x_i} \tag{4}$$

where $\overline{x}_i$ is the standardized variable, $x_i$ is variable, and n represents the total data sets (135). The $x_i$ was set as either *CC*, *CD*, or *FCCD*. However, the sector variable cannot be dealt with in the standardized method because it is a nominal data type. Therefore, the sector components (public, semi-public, and private) were considered in the ANN as factors. The names of the public, semi-public, and private were set as *PUB*, *SPUB*, and *PRI*, respectively. Depending on the data collection, the factors' values were changed to zero or one.

### 4.1.5. Data Augmentation

The ANN methods involve vast data to obtain a reliable forecasted model. However, the data sets were relatively small and may need to be improved. Pasini [28] innovated an "all frame" method to overcome small data issues. The method maximizes the small data by dividing them into N sets. The total data is divided into ten subsets. One is considered testing data, indicated as a blue element in Figure 4. At the same time, the remainder subsets represent training data, shown as white elements in Figure 2. The ten-training data set (ten groups) was generated by the relative positions of the testing data set, shown in Figure 4. The ten ANN models were developed with the same ANN structure based on the number of training data sets.

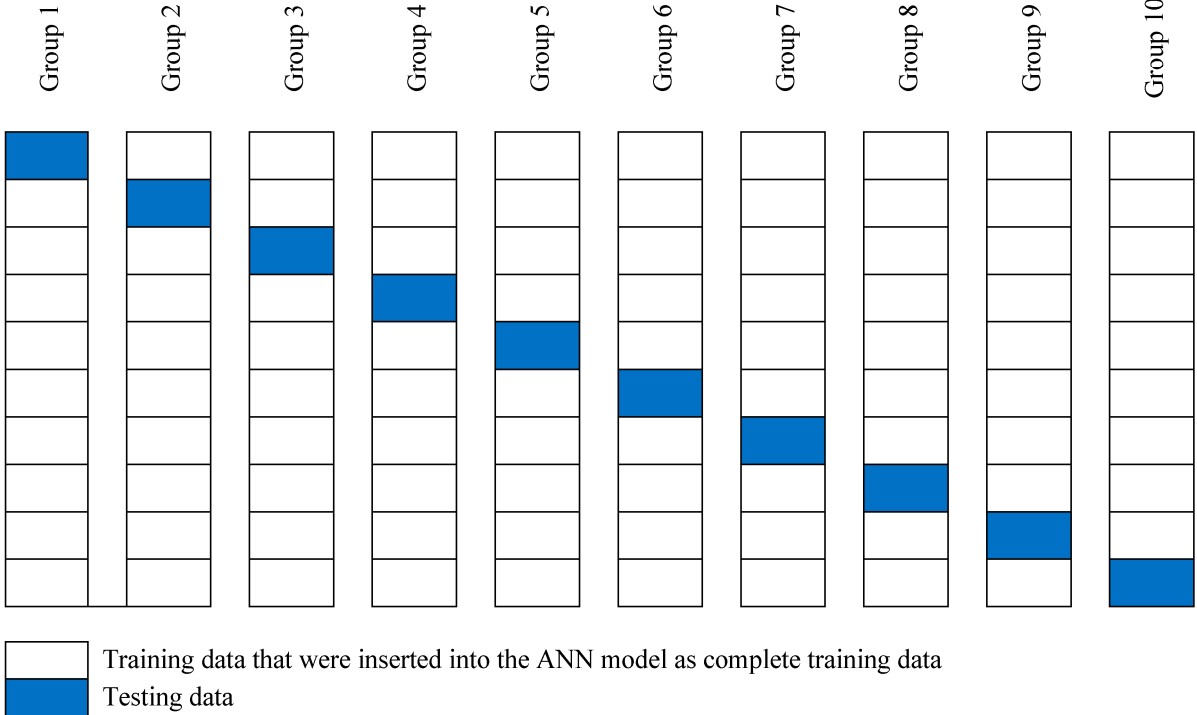

Training data that were inserted into the ANN model as complete training data
Testing data

**Figure 4.** The ten training data sets (groups).

### 4.2. ANN Model Development

The ANN model was developed by performing two analyses (first and second), as shown in the following section.

#### 4.2.1. First Analysis

The first analysis aims to detect the data that had significant differences between the observed and computed *FCCD*. It was accomplished by running ANN models and promoting the data, as detailed in the following section.

Running ANN Models

As illustrated in the main components of the ANN in the previous section, the input layer consists of two patterns of data: scale data ($\overline{CC}_i$, and $\overline{CD}_i$) and factors data (*PUB*, *SPUB*, and *PRI*). The $\overline{CC}_i$ and $\overline{CD}_i$ represent the standardized contract cost and standardized contract duration, respectively. The number of hidden layers was set to two, the greatest option in the SPSS-IBM program. In terms of the number of neurons in the hidden layers, they were set as (2m + 1) [34], where m is the number of the input layer neurons, which were five. Therefore, the number of neurons per hidden layer was eleven. Because the hyperbolic activation function is better than the sigmoid activation function [35], the activation function was set as hyperbolic. The output was set to one neuron ($\overline{FCCD}_i$). The ANN model' structure is shown in Figure 5.

Ten sets (groups) were considered for the ANN model based on the number of training data sets. Therefore, ten ANN models were developed. Each ANN model was run several times to monitor the Relative Error (RE). It should be close to a constant value to avoid overfitting [36,37]. The RE depended on the observed and computed $\overline{FCCD}_i$ Table 2 shows a portion of the input and output data used in ANN model 2.

Enhance the Used Data Accuracy

Some abnormal data have a detrimental effect on the accuracy of ANN models. There are tools to quality-check used data, such as residual error or APE. This paper identified abnormal data detection using an APE value of more than 35%. The modified training data

set was generated and will be used in the second analysis by deleting these data from the training data set. The APE can be computed using Equation (5).

$$APE_I = \frac{|FCCD_{obs-i} - FCCD_{com-i}|}{FCD_{obs-i}}$$

(5)

where $FCCD_{obs-i}$ and $FCCD_{com-i}$ are observed in the *FCCD*. $FCCD_{com-i}$ is computed from the *FCCD* based on the ANN model's outputs $\overline{FCCD}_i$. It should be noted that per the ANN model, there were several output values of the $\overline{FCCD}_i$ depending on the number of times the models were run after taking an average of the $\overline{FCD}_i$ values ($\overline{FCCD}_{average-i}$). The $FCCD_{com-i}$ is computed using Equation (3).

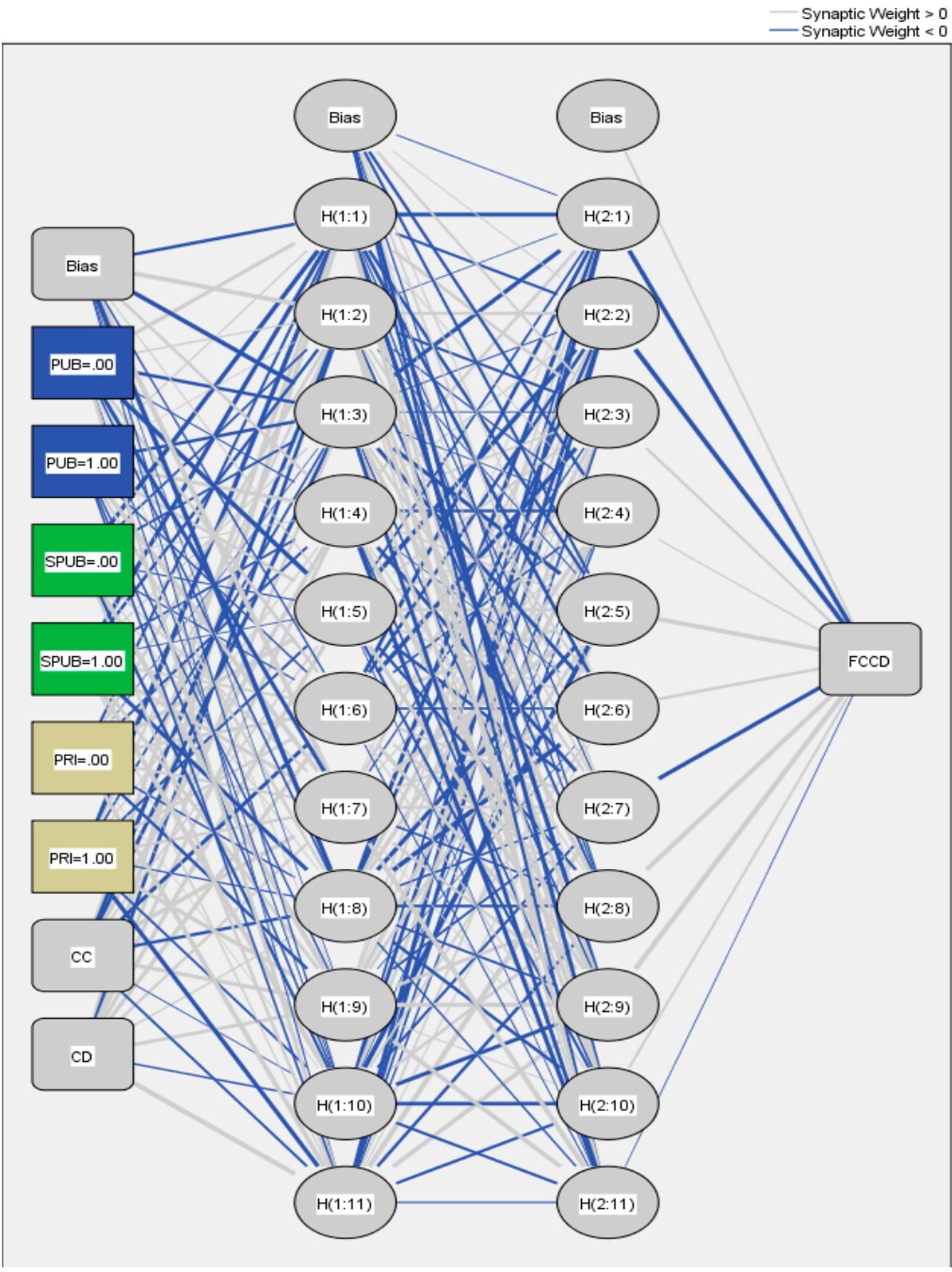

**Figure 5.** ANN model' structure.

**Table 2.** Illustrative example of input and output data used in ANN model 2.

| NO | Input | | | | | Output |
|---|---|---|---|---|---|---|
| | $\overline{CC}$ | $\overline{CD}$ | Public | Semi-Public | Private | $\overline{FCCD}$ |
| 1 | 20.66 | 13.01 | 0 | 1 | 0 | 14.90 |
| 2 | 33.08 | 0.05 | 0 | 1 | 0 | 0.04 |
| 3 | 20.20 | 12.68 | 0 | 1 | 0 | 10.86 |
| 4 | 27.01 | 14.46 | 0 | 1 | 0 | 17.03 |
| 5 | 35.23 | 15.22 | 0 | 1 | 0 | 13.85 |
| 6 | 22.57 | 14.77 | 0 | 1 | 0 | 15.72 |
| 7 | 21.58 | 16.81 | 0 | 1 | 0 | 17.60 |
| 8 | 32.86 | 15.92 | 0 | 1 | 0 | 15.82 |
| 9 | 34.51 | 14.39 | 0 | 1 | 0 | 17.46 |
| 10 | 29.65 | 14.95 | 0 | 1 | 0 | 15.77 |
| 11 | 24.43 | 14.77 | 0 | 1 | 0 | 15.04 |
| 12 | 30.00 | 14.95 | 0 | 1 | 0 | 16.25 |
| . | . | . | . | . | . | . |
| . | . | . | . | . | . | . |
| 122 | 33.68 | 16.38 | 1 | 0 | 0 | 16.36 |

### 4.2.2. Second Analysis

The ten ANN models—the same ANN model in the first analysis—were run several times based on the modified training data sets (ten modified data groups).

### *4.3. Evaluation Model*

The results of the ANN models that were carried out on the modified data were utilized to measure the accuracy of each model using the mean absolute percentage error (MAPE). It can be computed using Equation (6)

$$\text{MAPE} = \frac{1}{n_m} \sum_{i=1}^{n_m} \frac{|FCCD_{obs-i} - FCCD_{com-i}|}{FCD_{obs-i}} \qquad (6)$$

where $n_m$ is the number of modified data used in the ANN model. After that, the average of the ten MAPEs was computed. In addition, each model was tested with its test data to check the validity of the models for new data by measuring MAPE for each model.

## 5. Results and Discussions

Figure 6a,b shows the distribution frequency of the APE for the first and second analyses, respectively. Although the data for APE greater than 35% were deleted in the first analysis, there were some data for APE greater than 35% in the second analysis. In addition, the positive promotes data not only on essential data with high error (APE > 35%) but also on data with low error, as shown in Figure 6b. The trend of APE's frequency decreased with the APE value increase in the second analysis. In the first analysis, some data had a greater difference between the observed and computed *FCCD* than the observed *FCCD*, as shown in Figure 6a. This difference means that the data may contain a significant number of abnormal data.

Figure 5 shows the MAPE of the ten ANN models for the first and second analyses. The MAPE ranged from 27.5% to 32% for the first analysis. On the other hand, it ranged from 9.68% to 15.84% for the ANN models in the second analysis. The percentage removing data that had a significant difference between observed and computed *FCCD* (APE > 35%) was significant, with minimum and maximum values of 28% and 42%, respectively, as shown in Figure 7. Although the removing data percentage was high, the MAPE value decreased from 29.9% for the first analysis to 12.22% for the second analysis on average, which was close to the MAPE of the high accuracy models (MAPE ≤ 10%), as illustrated by [25,32,38].

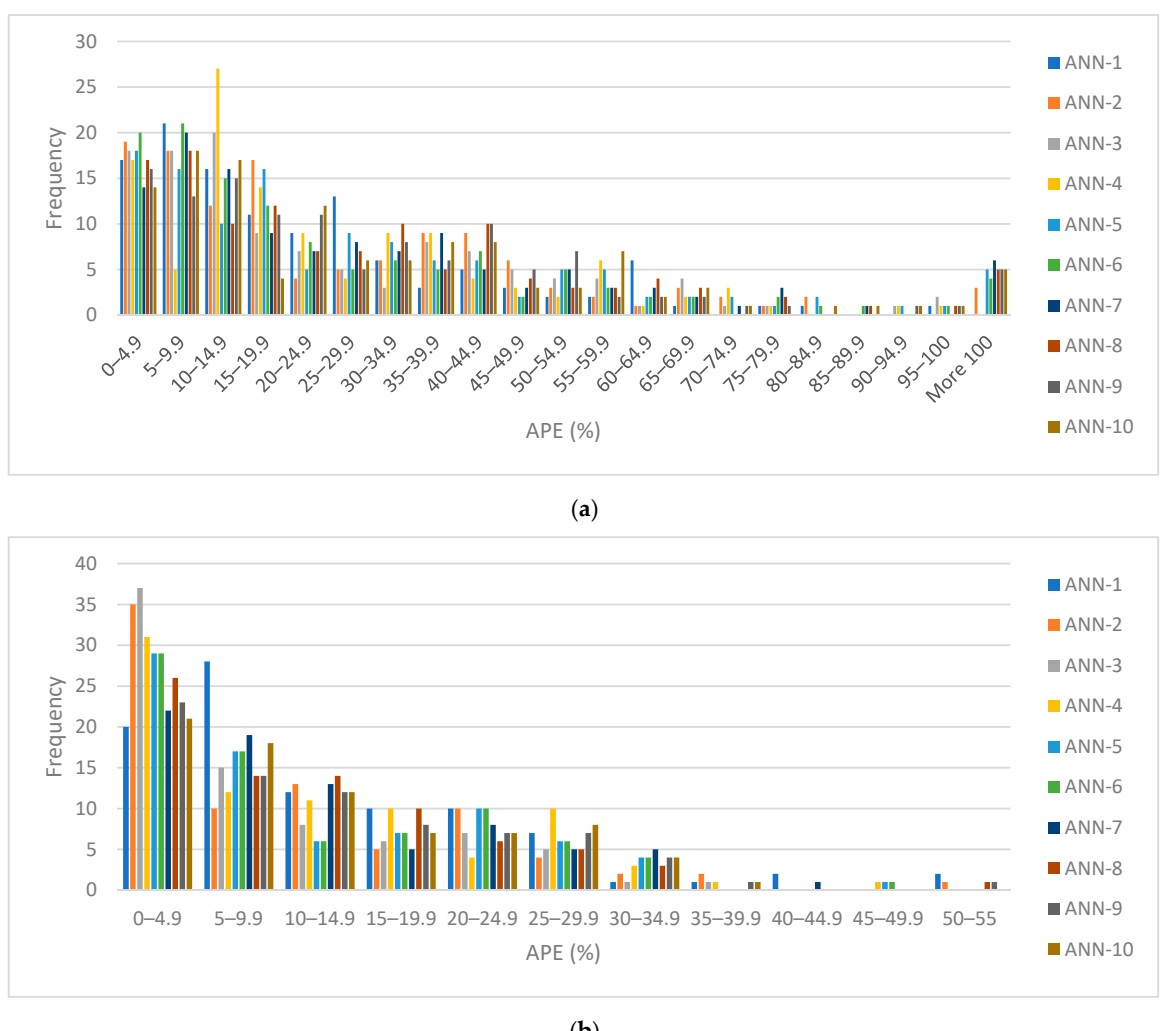

(**a**)

(**b**)

**Figure 6.** Frequency of APE of the first and second analysis for ten ANN models. (**a**) for first analysis (**b**) for second analysis.

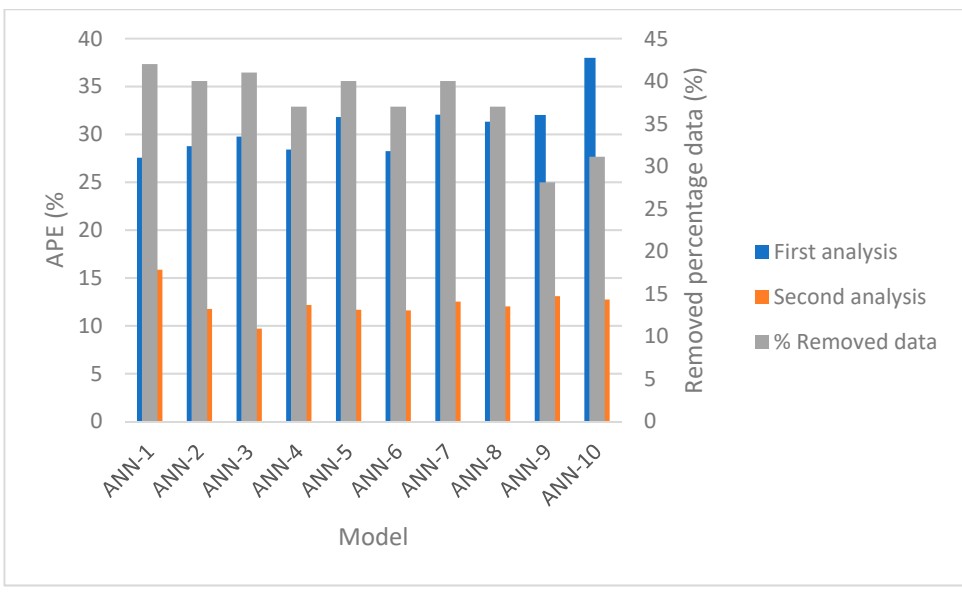

**Figure 7.** MAPE of the ten ANN models for the first and second analysis and the removing percentage.

Figure 8 shows the accumulative frequency of the APE; the data that had APE less than 10% varies from 48% to 65% among the ten ANN models. Moreover, the percentage of data that had APE less than 20% ranged from 75% to 81%. The difference ranges of the accumulative frequency decreased with increasing APE.

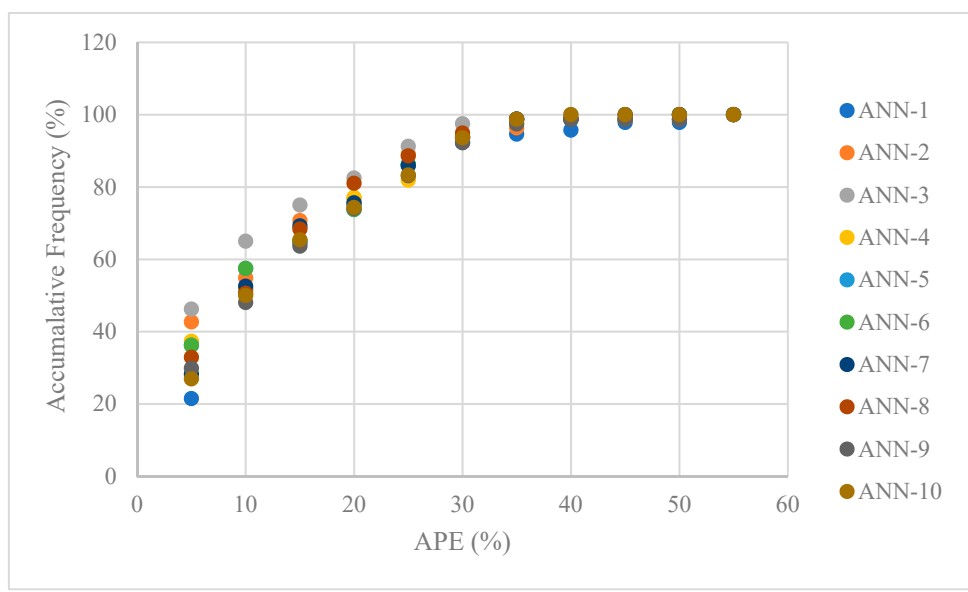

**Figure 8.** Accumulative frequencies versus APE of the ten ANN models of the second analysis.

The MAPE results of the ten models for testing data are shown in Figure 9. The MAPE value ranges from 3.47% (ANN-8) to 26.91% (ANN-3). The average MAPE of the ten models was 14.92%, less than the allowable standard value of 20% [27].

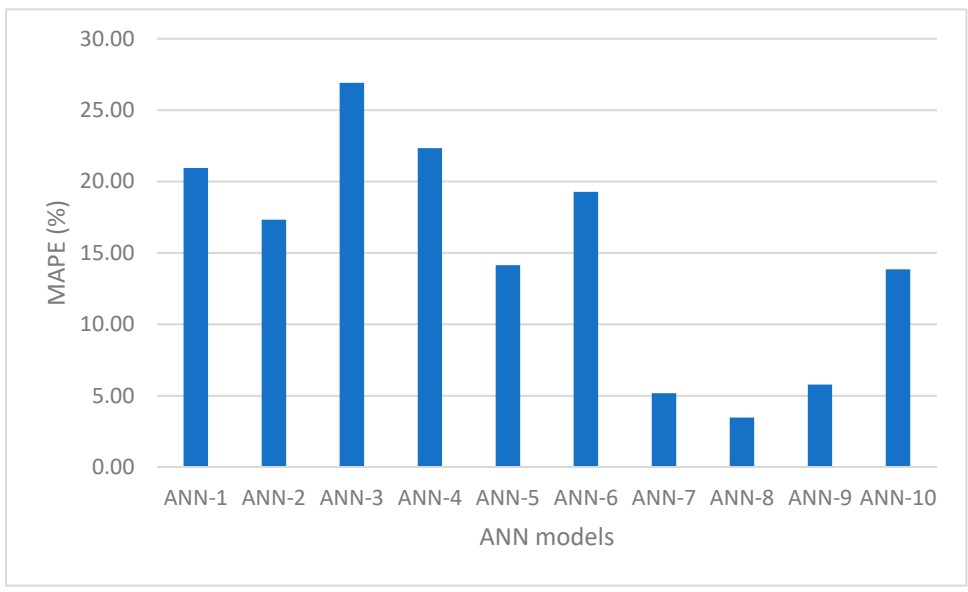

**Figure 9.** MAPE of ANN models for test data.

The MAPE of the three types of linear regression models (LR1, LR2, and LR3) was compared with the average MAPE of the ANN models in the first and second analyses. The LR1 represents linear regression performed on the 135 data sets without any transformed functions. LR2 and LR3 are linear regressions on the 135 data sets transformed by the square root and logarithmic function, respectively. The equations for LR1, LR2, and LR3 are shown in Table 3. The results indicate that the LR1 provides a high MAPE value of

143%, which indicates a low-accuracy model. On the other hand, the LR2 and LR3 models gave MAPEs of 43.6% and 39.23%, respectively, as shown in Figure 10. The results agreed with the results of [23,39]. They stated that the regression model of transformed data by the logarithmic function provides more accuracy than the other function. However, the three linear regressions had low accuracy due to the high value of the MAPE, as shown in Figure 10. The average value of the MAPE for the first and second analyses was 29.9% and 12.22%, respectively. They are lower than the LR3 by 9.3% and 27.03%, respectively. The paper's contribution is to increase the accuracy of the ANN model in predicting the *FCCD* based on relatively small data by using the three methods of preparation (standardize, augment, and promote data).

**Table 3.** Description of the different regression models.

| Model | Regression Formula |
|---|---|
| LR1 | $FCD = -5.898 + 1.227CD + 0.085CC + 7.532Sector$ |
| LR2 | $FCD = \left(-5.898 + 1.227\sqrt{CD} + 0.085\sqrt{CC} + 7.532Sector\right)^2$ |
| LR3 | $FCD = Exp(-5.898 + 1.227\ln CD + 0.085\ln CC + 7.532Sector)$ |

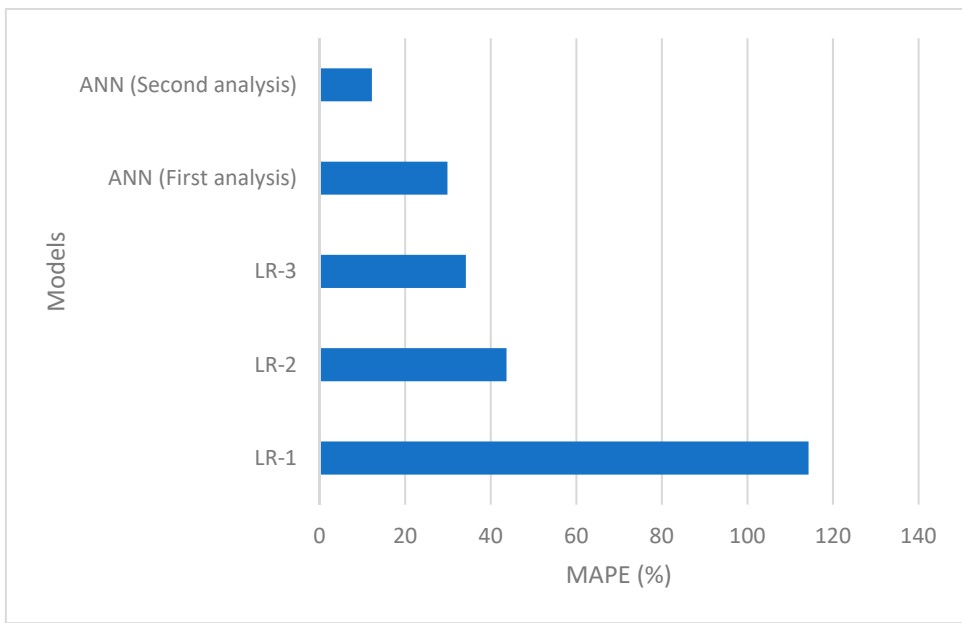

**Figure 10.** MAPE of the three linear regression and the average of MAPE of the ANN models for the first and second analyses.

To study the performance of *FCCD*, the ANN-5 was considered because the MAPE of the model for training and testing stages for the second analysis provides an average value among the models. Figure 11 shows the variation of the *FCCD* with *CD* for public, semi-public, and private. The *FCCD* value of the public and semi-public has been overestimated. However, the *FCCD* value for private is underestimated. To compare these results with a study by Skitmore and Ng [23] for residential Australian construction projects, the type of contract is a lump sum, similar to Saudi construction. The Skitmore and Ng curve was between the public and semi-public curves for *CD* for more than ten months. On the other hand, the previous study curve was close to the semi-public curve for *CD* for less than ten months.

The accuracy of ANN models was compared with different studies. The CBR model developed by Jin et al. [22] considered geometry, building information, foundation system, subsoil condition, and roof type. The average APE was 5.74%. The value was smaller than the MAPE of the ANN model. The CBR model was used for building projects, while the

ANN model utilized projects such as building, highway, electric, and mechanic projects. Therefore, the margin of error may be increased. To compare the accuracy of the ANN model with the EVM developed by [21], the EVM provides a MAPE value of 12.96%, which is close to the MAPE of the ANN model (12.22%) for this paper. On the other hand, the MAPE of the ANN model performed by Pewdum et al. [15] was 6.2% on average. It was less than the developed ANN' MAPE. It is assumed that the Pewdum ANN model's input parameters represent the working start date, CD, % actual completion, evaluating date, and % planned completion. These parameters were closely related to the *FCCD*. However, some parameters were not available in the pre-tendering phase.

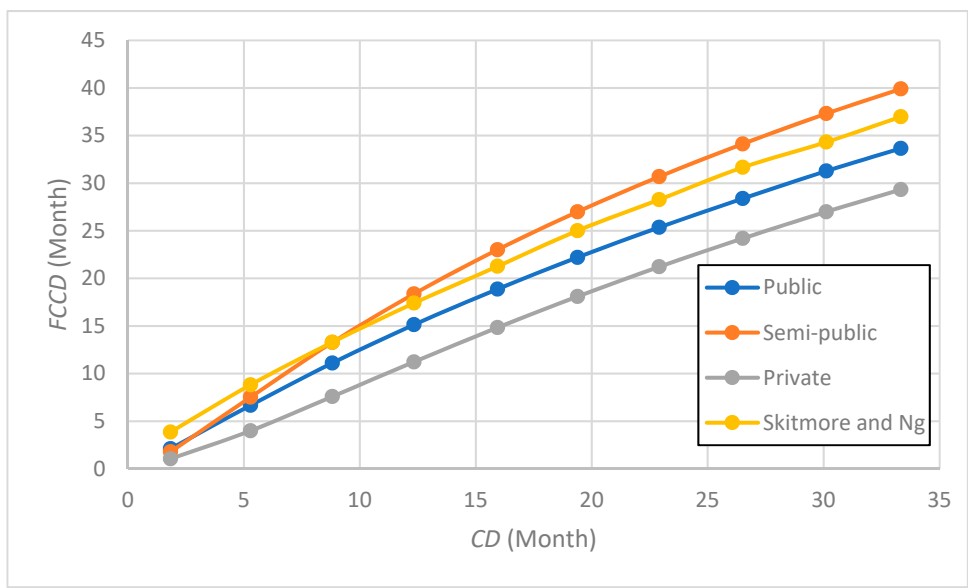

**Figure 11.** *FCCD* vs. *CD* for three types of sectors with the curve obtained by Skitmore and Ng for *CC* of AUS$ 100 million.

## 6. Conclusions

Early-stage estimation of the final construction contract duration is crucial for the progress and success of a project. The 135 data sets from Saudi projects were used to create the ANN models in the paper. The ANN model's development consists of three phases. The first phase was to collect the data, process the data using Zavadskas and Turskis' logarithmic standardization method, augment the data using the method introduced by [33], and generate ten training data sets. In the first analysis, ten ANN models were developed to detect the data with a significant value of APE. These data were then deleted to obtain the modified ten training data sets used in the second analysis. The main findings revealed that the average MAPE of the ANN models in the second analysis was 12.22%, and the model accuracy was high to good. In addition, the ANN model provides better performance than the LR model, especially the linear model that transforms data by a logarithmic function. The paper supports the research using comparatively little data and neural network integration.

**Author Contributions:** Conceptualization, A.M.A. and K.S.A.-G.; data curation, N.M.A. and A.A.A.; formal analysis, N.M.A. and K.S.A.-G.; funding acquisition, A.M.A. and K.S.A.-G.; investigation, A.M.A., N.M.A. and K.S.A.-G.; methodology, N.M.A., K.S.A.-G. and A.A.A.; project administration, A.M.A., K.S.A.-G. and A.S.A.; resources, A.M.A. and K.S.A.-G.; software, N.M.A.; supervision, A.M.A. and K.S.A.-G.; validation, N.M.A., K.S.A.-G. and A.A.A.; visualization, A.M.A., N.M.A., K.S.A.-G. and A.S.A.; Writing—original draft preparation, A.M.A., N.M.A., A.A.A. and K.S.A.-G.; Writing—review and editing A.M.A., N.M.A., K.S.A.-G. and A.A.A. All authors have read and agreed to the published version of the manuscript.

**Funding:** The authors extend their appreciation to the Deputyship for Research and Innovation, Ministry of Education in Saudi Arabia, for funding this research work through project no. (IFKSUOR3-380-2).

**Institutional Review Board Statement:** Not applicable.

**Informed Consent Statement:** Not applicable.

**Data Availability Statement:** The raw data supporting the findings of this paper are available on request from the corresponding author.

**Conflicts of Interest:** The authors declare no conflict of interest.

## Appendix A

**Table A1.** Table Data of 136 projects.

| *CC* (SAR) | *CD* (Month) | *FCCD* (Month) | Public | Semi-Public | Private |
|---|---|---|---|---|---|
| 1,500,000 | 138.30 | 134.83 | 0.00 | 1.00 | 0.00 |
| 68,460,000 | 33.00 | 33.47 | 0.00 | 1.00 | 0.00 |
| 51,438,229 | 23.97 | 77.83 | 0.00 | 1.00 | 0.00 |
| 15,499,528 | 23.60 | 75.87 | 0.00 | 1.00 | 0.00 |
| 11,739,569 | 10.13 | 34.33 | 0.00 | 1.00 | 0.00 |
| 42,791,190 | 30.42 | 30.33 | 0.00 | 1.00 | 0.00 |
| 27,490,435 | 33.80 | 37.87 | 0.00 | 1.00 | 0.00 |
| 3,889,500 | 14.77 | 76.10 | 0.00 | 1.00 | 0.00 |
| 62,974,880 | 23.60 | 23.60 | 0.00 | 1.00 | 0.00 |
| 19,265,430 | 20.47 | 36.67 | 0.00 | 1.00 | 0.00 |
| 27,058,829 | 14.77 | 28.77 | 0.00 | 1.00 | 0.00 |
| 21,520,394 | 24.00 | 23.93 | 0.00 | 1.00 | 0.00 |
| 79,890,345 | 20.17 | 28.23 | 0.00 | 1.00 | 0.00 |
| 73,802,382 | 29.50 | 48.13 | 0.00 | 1.00 | 0.00 |
| 23,106,764 | 17.23 | 44.80 | 0.00 | 1.00 | 0.00 |
| 152,539,013 | 1.00 | 1.00 | 0.00 | 1.00 | 0.00 |
| 21,520,394 | 16.03 | 15.97 | 0.00 | 1.00 | 0.00 |
| 60,606,426 | 23.63 | 77.10 | 0.00 | 1.00 | 0.00 |
| 211,300,000 | 27.93 | 34.23 | 0.00 | 1.00 | 0.00 |
| 30,851,600 | 25.33 | 55.23 | 0.00 | 1.00 | 0.00 |
| 26,562,763 | 39.50 | 89.23 | 0.00 | 1.00 | 0.00 |
| 147,451,259 | 32.53 | 56.70 | 0.00 | 1.00 | 0.00 |
| 189,342,875 | 23.27 | 86.10 | 0.00 | 1.00 | 0.00 |
| 90,562,500 | 26.30 | 56.00 | 0.00 | 1.00 | 0.00 |
| 40,948,000 | 25.30 | 46.43 | 0.00 | 1.00 | 0.00 |
| 95,500,000 | 26.30 | 63.23 | 0.00 | 1.00 | 0.00 |
| 6,535,200 | 12.10 | 12.10 | 0.00 | 1.00 | 0.00 |
| 57,382,472 | 36.50 | 94.60 | 0.00 | 1.00 | 0.00 |
| 101,852,500 | 24.00 | 84.37 | 0.00 | 1.00 | 0.00 |
| 103,084,608 | 29.50 | 78.10 | 0.00 | 1.00 | 0.00 |

**Table A1.** *Cont.*

| CC (SAR) | CD (Month) | FCCD (Month) | Public | Semi-Public | Private |
|---|---|---|---|---|---|
| 110,906,432 | 29.50 | 67.90 | 0.00 | 1.00 | 0.00 |
| 35,840,000 | 17.67 | 52.83 | 0.00 | 1.00 | 0.00 |
| 39,473,010 | 22.97 | 45.07 | 0.00 | 1.00 | 0.00 |
| 42,585,725 | 22.97 | 66.80 | 0.00 | 1.00 | 0.00 |
| 63,554,955 | 22.97 | 82.23 | 0.00 | 1.00 | 0.00 |
| 26,787,515 | 24.33 | 37.80 | 0.00 | 1.00 | 0.00 |
| 1,976,113 | 24.33 | 37.80 | 0.00 | 1.00 | 0.00 |
| 22,321,935 | 24.33 | 37.80 | 0.00 | 1.00 | 0.00 |
| 4,827,143 | 12.20 | 17.80 | 0.00 | 1.00 | 0.00 |
| 4,908,491 | 5.90 | 5.43 | 0.00 | 0.00 | 0.00 |
| 8,705,670 | 3.03 | 6.03 | 0.00 | 1.00 | 0.00 |
| 60,170,000 | 24.33 | 31.83 | 0.00 | 1.00 | 0.00 |
| 95,154,746 | 30.42 | 35.33 | 0.00 | 1.00 | 0.00 |
| 51,711,016 | 24.33 | 72.80 | 0.00 | 1.00 | 0.00 |
| 52,922,740 | 24.33 | 72.80 | 0.00 | 1.00 | 0.00 |
| 45,947,586 | 24.33 | 42.63 | 0.00 | 1.00 | 0.00 |
| 5,530,000 | 35.10 | 63.33 | 0.00 | 1.00 | 0.00 |
| 17,202,401 | 18.23 | 46.73 | 0.00 | 1.00 | 0.00 |
| 14,486,980 | 11.20 | 16.50 | 0.00 | 1.00 | 0.00 |
| 30,771,774 | 15.13 | 24.23 | 0.00 | 1.00 | 0.00 |
| 651,000,000 | 12.00 | 13.00 | 1.00 | 0.00 | 0.00 |
| 103,040,000 | 36.00 | 39.00 | 1.00 | 0.00 | 0.00 |
| 86,400,000 | 3.00 | 32.00 | 1.00 | 0.00 | 0.00 |
| 12,000,000 | 12.00 | 15.00 | 1.00 | 0.00 | 0.00 |
| 82,351,564 | 6.50 | 30.42 | 1.00 | 0.00 | 0.00 |
| 90,060,950 | 16.30 | 33.33 | 1.00 | 0.00 | 0.00 |
| 8,850,000 | 14.00 | 9.00 | 1.00 | 0.00 | 0.00 |
| 21,491,170 | 30.00 | 29.00 | 1.00 | 0.00 | 0.00 |
| 13,957,188 | 11.00 | 11.00 | 1.00 | 0.00 | 0.00 |
| 72,990 | 30.00 | 28.00 | 1.00 | 0.00 | 0.00 |
| 207,475 | 2.00 | 2.00 | 1.00 | 0.00 | 0.00 |
| 140,726,856 | 27.00 | 41.00 | 1.00 | 0.00 | 0.00 |
| 22,259,958 | 13.00 | 22.00 | 1.00 | 0.00 | 0.00 |
| 154,600 | 2.00 | 1.90 | 1.00 | 0.00 | 0.00 |
| 39,704,154 | 42.00 | 48.00 | 1.00 | 0.00 | 0.00 |
| 3,454,386 | 12.00 | 12.00 | 1.00 | 0.00 | 0.00 |
| 74,700 | 1.00 | 0.73 | 1.00 | 0.00 | 0.00 |
| 213,500 | 3.00 | 2.60 | 1.00 | 0.00 | 0.00 |
| 299,250 | 1.00 | 1.17 | 1.00 | 0.00 | 0.00 |
| 32,142,222 | 18.00 | 34.00 | 1.00 | 0.00 | 0.00 |



**Table A1.** *Cont.*

| CC (SAR) | CD (Month) | FCCD (Month) | Public | Semi-Public | Private |
|---|---|---|---|---|---|
| 289,217 | 1.00 | 0.37 | 1.00 | 0.00 | 0.00 |
| 92,694 | 1.50 | 0.80 | 1.00 | 0.00 | 0.00 |
| 18,200 | 1.00 | 0.70 | 1.00 | 0.00 | 0.00 |
| 352,685 | 3.00 | 2.83 | 1.00 | 0.00 | 0.00 |
| 226,653 | 4.00 | 12.00 | 1.00 | 0.00 | 0.00 |
| 105,000 | 1.00 | 0.90 | 1.00 | 0.00 | 0.00 |
| 87,034 | 1.50 | 0.90 | 1.00 | 0.00 | 0.00 |
| 496,000 | 0.50 | 1.03 | 1.00 | 0.00 | 0.00 |
| 480,000 | 0.50 | 0.37 | 1.00 | 0.00 | 0.00 |
| 256,800 | 1.50 | 2.50 | 1.00 | 0.00 | 0.00 |
| 574,590 | 6.00 | 8.93 | 1.00 | 0.00 | 0.00 |
| 690,450 | 2.00 | 1.70 | 1.00 | 0.00 | 0.00 |
| 488,775 | 4.00 | 4.00 | 1.00 | 0.00 | 0.00 |
| 491,400 | 4.00 | 4.00 | 1.00 | 0.00 | 0.00 |
| 481,950 | 6.00 | 13.00 | 1.00 | 0.00 | 0.00 |
| 489,510 | 3.00 | 2.70 | 1.00 | 0.00 | 0.00 |
| 444,000 | 3.00 | 3.67 | 1.00 | 0.00 | 0.00 |
| 11,835,120 | 22.00 | 20.00 | 1.00 | 0.00 | 0.00 |
| 232,281 | 3.00 | 10.93 | 1.00 | 0.00 | 0.00 |
| 296,100 | 3.00 | 2.97 | 1.00 | 0.00 | 0.00 |
| 287,000 | 2.00 | 2.00 | 1.00 | 0.00 | 0.00 |
| 285,100 | 2.00 | 2.00 | 1.00 | 0.00 | 0.00 |
| 296,500 | 1.00 | 1.00 | 1.00 | 0.00 | 0.00 |
| 297,000 | 1.00 | 1.00 | 1.00 | 0.00 | 0.00 |
| 285,000 | 1.00 | 1.00 | 1.00 | 0.00 | 0.00 |
| 247,800 | 2.00 | 2.00 | 1.00 | 0.00 | 0.00 |
| 299,810 | 2.00 | 2.00 | 1.00 | 0.00 | 0.00 |
| 299,907 | 2.00 | 2.00 | 1.00 | 0.00 | 0.00 |
| 96,200 | 1.50 | 1.50 | 1.00 | 0.00 | 0.00 |
| 295,750 | 2.00 | 2.00 | 1.00 | 0.00 | 0.00 |
| 230,000 | 2.00 | 2.00 | 1.00 | 0.00 | 0.00 |
| 299,700 | 2.00 | 2.00 | 1.00 | 0.00 | 0.00 |
| 247,300 | 6.00 | 6.00 | 1.00 | 0.00 | 0.00 |
| 76,000 | 1.50 | 1.50 | 1.00 | 0.00 | 0.00 |
| 469,000 | 3.00 | 3.00 | 1.00 | 0.00 | 0.00 |
| 33,500,000 | 6.00 | 6.00 | 1.00 | 0.00 | 0.00 |
| 6,974,000 | 6.00 | 7.50 | 1.00 | 0.00 | 0.00 |
| 15,996,979 | 6.00 | 6.00 | 1.00 | 0.00 | 0.00 |
| 499,000 | 3.00 | 3.00 | 1.00 | 0.00 | 0.00 |
| 478,000 | 3.00 | 3.00 | 1.00 | 0.00 | 0.00 |

**Table A1.** *Cont.*

| CC (SAR) | CD (Month) | FCCD (Month) | Public | Semi-Public | Private |
|---|---|---|---|---|---|
| 451,000 | 3.00 | 3.00 | 1.00 | 0.00 | 0.00 |
| 50,000,000 | 8.00 | 8.13 | 0.00 | 0.00 | 1.00 |
| 190,000,000 | 12.47 | 14.17 | 0.00 | 0.00 | 1.00 |
| 20,000,000 | 6.00 | 24.33 | 0.00 | 0.00 | 1.00 |
| 35,000,000 | 11.93 | 11.17 | 0.00 | 0.00 | 1.00 |
| 11,000,000 | 12.00 | 5.00 | 0.00 | 0.00 | 1.00 |
| 291,000 | 1.00 | 1.00 | 1.00 | 0.00 | 0.00 |
| 259,500 | 1.00 | 1.00 | 1.00 | 0.00 | 0.00 |
| 260,000 | 1.00 | 1.00 | 1.00 | 0.00 | 0.00 |
| 295,576 | 0.93 | 0.93 | 1.00 | 0.00 | 0.00 |
| 259,259 | 0.93 | 0.93 | 1.00 | 0.00 | 0.00 |
| 61,380 | 1.00 | 1.00 | 1.00 | 0.00 | 0.00 |
| 35,500 | 0.50 | 0.50 | 1.00 | 0.00 | 0.00 |
| 50,000 | 0.47 | 0.47 | 1.00 | 0.00 | 0.00 |
| 221,250 | 4.00 | 4.00 | 1.00 | 0.00 | 0.00 |
| 90,000 | 1.00 | 1.00 | 1.00 | 0.00 | 0.00 |
| 119,333 | 3.00 | 3.00 | 1.00 | 0.00 | 0.00 |
| 593,295 | 6.00 | 6.00 | 1.00 | 0.00 | 0.00 |
| 120,000 | 1.00 | 1.00 | 1.00 | 0.00 | 0.00 |
| 390,000 | 2.00 | 2.70 | 1.00 | 0.00 | 0.00 |
| 64,000,000 | 30.50 | 36.50 | 1.00 | 0.00 | 0.00 |
| 49,998,494 | 18.25 | 19.00 | 1.00 | 0.00 | 0.00 |
| 24,756,246 | 18.25 | 19.00 | 1.00 | 0.00 | 0.00 |
| 173,000,000 | 14.00 | 68.00 | 1.00 | 0.00 | 0.00 |
| 470,000,000 | 36.00 | 146.00 | 1.00 | 0.00 | 0.00 |

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
