# Peer review of "Artificial Neural Network Model to Predict Final Construction Contract Duration"

_applsci, doi:10.3390/app13148078_

Round 1

Reviewer 1 Report

In general, the direction of research chosen by the authors is interesting and relevant. However, the article still needs revision and improvement.

Some points that have to be improved.

1. The authors need to show what is the novelty of this study in comparison with analogues? This information should be added to the abstract and introduction.

2. Lines 64 - 77. I think it should be moved to the Methods section.

3. The literature review does not correspond to the stated study topic. It is necessary to analyze the experience to use ML algorithms to predict the duration of a construction project.

4. I don't understand the meaning of section 3.

5. The data collection phase is poorly described. It is necessary to describe which variables were used as inputs, as well as justify their choice.

6. I recommend that authors add a portion of the input and input data as an example in tabular form.

7. Table 1. Please describe what the information in the second column means.

8. The authors claim that the proposed model can be used to predict the duration of a construction project. In such a case, the article should show how the trained model predicts the value of "construction project duration" based on new inputs that the model has not yet "seen". At the same time, the authors ended the article with a simple comparison of the performance of the models. I think this is not enough.

9. The discussion section. The problem of predicting construction project duration using various ML tools has already been repeatedly considered by scientists. Accordingly, the "Discussion" section should contain a comparison of the results of this article and similar studies by other authors. The authors need to prove the superiority and scientific novelty of this study.

Author Response

The authors thank you for your valuable comments to improve the paper's quality. Please find the attached file is a tabulated answer to your comments.

Reviewer 2 Report

The paper is good and can be accepted in the current form. Mnay thanks for authors 

good 

Author Response

The authors would like to thank you for your time to review the paper. Much appreciated.

Reviewer 3 Report

The paper presents a study on the development of ML models to forecast civil project duration. The paper is well written and presented, and in my opinion could be accepted for publication. Few aspects should be improved before of the final acceptance:

- The introduction and the state of the art are well written and motivated, even if some aspects could be added, such as the use of ML models in the field of civil engineering, such as vulma. Some works are reported in 10.1016/j.engfailanal.2023.107237. Please check the fields of application and references therein

- The part of ANN should be enlarged, since a very short description is provided

- Figure 1 is not clear. In addition I see 2 times the evaluation of the model. Is this an error?

- Some doubts exist about the data. What are the parameters that authors considered for the training? And what are the parameters useful for the scope of the study, which is to reduce delay in projects?

- Figure 2 should be represented according to the data by authors. It seems too generic and it is not adequate for the paper.

-  at page 7 we have: Promote the used data. What did authors mean? Is it a title?

- In addition to show the results of ANN, authors should report some information about the physics under the model, which allows to achieve the aim of the study.

Minor modifications are required

Author Response

(The authors gave the same response as above.)

Round 2

Reviewer 1 Report

I recommend publishing the article in present form 

Author Response

The authors would like to thank the reviewer for their time to review the paper. Much appreciated.

Reviewer 3 Report

Comments are not corresponding to the manuscript revision. Please, check the document 

Author Response

The authors would like to thank the reviewer for their time to review the paper. We are much appreciated. The attached file is our answer to your comments.

Round 3

Reviewer 3 Report

The paper is now ready to be published